# Experimental and Comparative Study of Rotor Vibrations of Permanent Magnet Machines with Two Different Fractional Pole/Slot Combinations

**Tae-Kyoung Bang** [1], **Kyung-Hun Shin** [2], **Jeong-In Lee** [1], **Hoon-Ki Lee** [1], **Han-Wook Cho** [3] **and Jang-Young Choi** [1,*]

[1] Department of Electrical Engineering, Chungnam National University, Daejeon 305-764, Korea; bangtk77@cnu.ac.kr (T.-K.B.); lji477@cnu.ac.kr (J.-I.L.); lhk1109@cnu.ac.kr (H.-K.L.)

[2] Department of Power System Engineering, Chonnam National University, 50 Daehak-ro, Yeosu 59626, Korea; kshin@jnu.ac.kr

[3] Department of Electric. Electro. and Communication Eng. Edu., Chungnam National University, Daejeon 305-764, Korea; hwcho@cnu.ac.kr

\* Correspondence: choi_jy@cnu.ac.kr

**Abstract:** This study deals with the noise, vibration, and harshness (NVH) characteristic analysis of permanent magnet synchronous motors (PMSMs) for electrical machines, such as electrically driven tools that are used in industries. An improved NVH design is needed for application to industrial tools. In general, the electromagnetic NVH characteristics of PMSMs are classified into electromagnetic excitation sources, such as total harmonic distortion of EMF, torque pulsation, magnetic pull force, and unbalanced magnetic force (UMF). This study compares the vibration and noise generated by fractional pole/slot combinations. In PMSMs with fractional pole/slot combinations, UMF is an important NVH source. PMSMs generate UMF because of armature reaction fields based on the pole/slot combinations and harmonics of magnetic flux density. UMF was derived using the finite element method, and the rotor vibration analysis was performed using electromagnetic mechanical coupling analysis. The analysis results and the effect of electromagnetic excitation characteristics on the rotor vibration of the PMSMs were compared and analyzed.

**Keywords:** PMSMs; mechanical stress analysis; electromagnetic–mechanical analysis

## 1. Introduction

In recent times, permanent magnet synchronous motors (PMSMs) have been widely used in industries because of their various advantages, such as high-speed operation, high efficiency, and compact design [1–4]. However, PMSMs have the disadvantages of noise and vibration caused by high magnetic energy during the interaction between stator and rotor magnets; these drawbacks significantly affect machine performance. Moreover, vibration and noise cause eccentricity, bearing faults, and misalignment of PMSMs [5–10]. Therefore, it is important to determine electromagnetic vibration sources that cause vibration and noise. Consequently, torque pulsation should be considered during the design stage.

Electromagnetic vibration sources can be divided into two types: torque pulsation and electromagnetic force [5–8]. Torque pulsation is classified into cogging torque and torque ripple. Cogging torque is the torsional force that is generated by the interaction between the magnetic flux and the slotting effect that prevents the rotors from rotating in a specific direction. Torque ripple is the load pulsation generated by the interaction between the input current and the back-EMF harmonics. Previous studies have reported that vibration characteristics can be improved by decreasing torque pulsation; these are related to the noise characteristic of PMSMs [5–8]. Electromagnetic force is classified into unbalanced

electromagnetic force (UMF) and magnetic pull force (MPF). MPF is caused by the distribution of magnetic flux density that is the product of the winding pattern and the slotting effect. It implies the distribution of the magnetic force and, thus, should be symmetrical to prevent electrical machine failure. It affects electrical machines by the magnitude of the force generated, not by force distribution. Moreover, various studies emphasizing the symmetry of MPF have been published [9,10]. In contrast, UMF is caused by changing the magnetic flux density, which is the main cause of vibration, noise, and rotor eccentricity in induction motors. There are a few studies on the UMF of permanent magnet (PM) machines. L.J. Wu and Z.Q. Zhu analyzed the harmonic orders of UMF using an analytical method and experimental results [11]. Moreover, Z.Q. Zhu analyzed the UMF generated by models based on winding patterns, with equal specifications and changes in the current waveform according to the shape of the machines [12]. Based on these studies, the asymmetry of MPF affects UMF and increases the amplitude of UMF. Therefore, D.Y. Kim conducted a study on vibration reduction based on pole/slot combinations for the reduction of electromagnetic vibration source [13]. D. Torregrossa, Z. Huang, and S. Das studied the analysis method of NVH using multiphysics analysis [14–18]. However, they did not analyze which electromagnetic sources could affect NVH or were not considered multiphysics analysis. Therefore, to analyze the influence of each source, we propose a design of PMSMs with fractional pole/slot combinations of equivalent characteristic performance; we derive the dominant model of pole/slot combination for each electromagnetic vibration source based on finite element analysis and compare the results of the electromagnetic characteristic analysis.

The derived pole/slot combinations of the models are 8-poles/9-slots and 8-poles/12-slots. The electromagnetic analysis and prototype models used in the NVH analysis are shown in Figure 1. The dominant components of the 9- and 12-slot models are torque pulsation and electromagnetic force, respectively. Thus, we have analyzed the influence of each vibration source by conducting experiments and electromechanical coupled analysis using finite element method (FEM) and then compared the results with two different fractional pole/slot combinations.

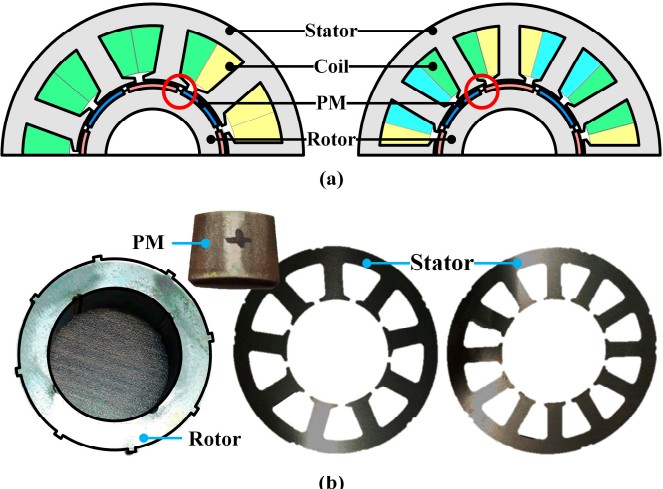

**Figure 1.** The analysis model of permanent magnet machines with two different pole/slot combinations: (**a**) electromagnetic analysis model and (**b**) prototype model.

## 2. Comparative Electromagnetic Analysis

### 2.1. Back-EMF and Torque

In the case of PM machines, a decrease in total harmonic distortion (THD) of back-EMF leads to high torque density because the input current decreases to generate an equal torque. In addition, it is not proportional to the size of the THD because it affects the torque ripple. Therefore, a PM machine

should be designed to accommodate sinusoidal waveforms. The phase back-EMF and the current are described in Equation (1).

$$e_{phase} = \omega_r \sum_{n=1,odd}^{\infty} K_n \sin(np\omega_r t)$$
$$i_{phase} = I_1 \sin(p\omega_r t)$$
(1)

where $\omega_r$ is the rotational speed, $t$ is the time, $K_n$ is the back-EMF coefficient, $I_1$ is the peak of current, and $np$ indicates the $n$th-order harmonic and the number of pole pairs, respectively. The electromagnetic torque is expressed as the product of the back-EMF and the current (Equation (2)).

$$T = \frac{3}{2}K_1 I_1$$
$$+ \frac{1}{2}I_1 \sum_{n=1,odd}^{\infty} K_n[1 + 2\cos\{(n-1) \cdot 2\pi/3\}] \times \cos(n-1)p\omega_r t$$
$$+ \frac{1}{2}I_1 \sum_{n=1,odd}^{\infty} K_n[1 + 2\cos\{(n+1) \cdot 2\pi/3\}] \times \cos(n+1)p\omega_r t$$
(2)

Here, the first term represents the average electromagnetic torque, and the second and third terms represent the torque ripple produced by the THD of the back-EMF and the current, respectively. Thus, the amplitude of the harmonics of the back-EMF and the current affects the magnitude of the torque ripple. Figure 2a,b compare the back-EMF analysis results of the two models and indicate that the 9-slot model has better THD than the 12-slot model. The THD of the 9- and 12-slot models is 20.10% and 16.7726, respectively, and it can be seen that the harmonics of the 9-slot model is larger than that of the 12-slot model.

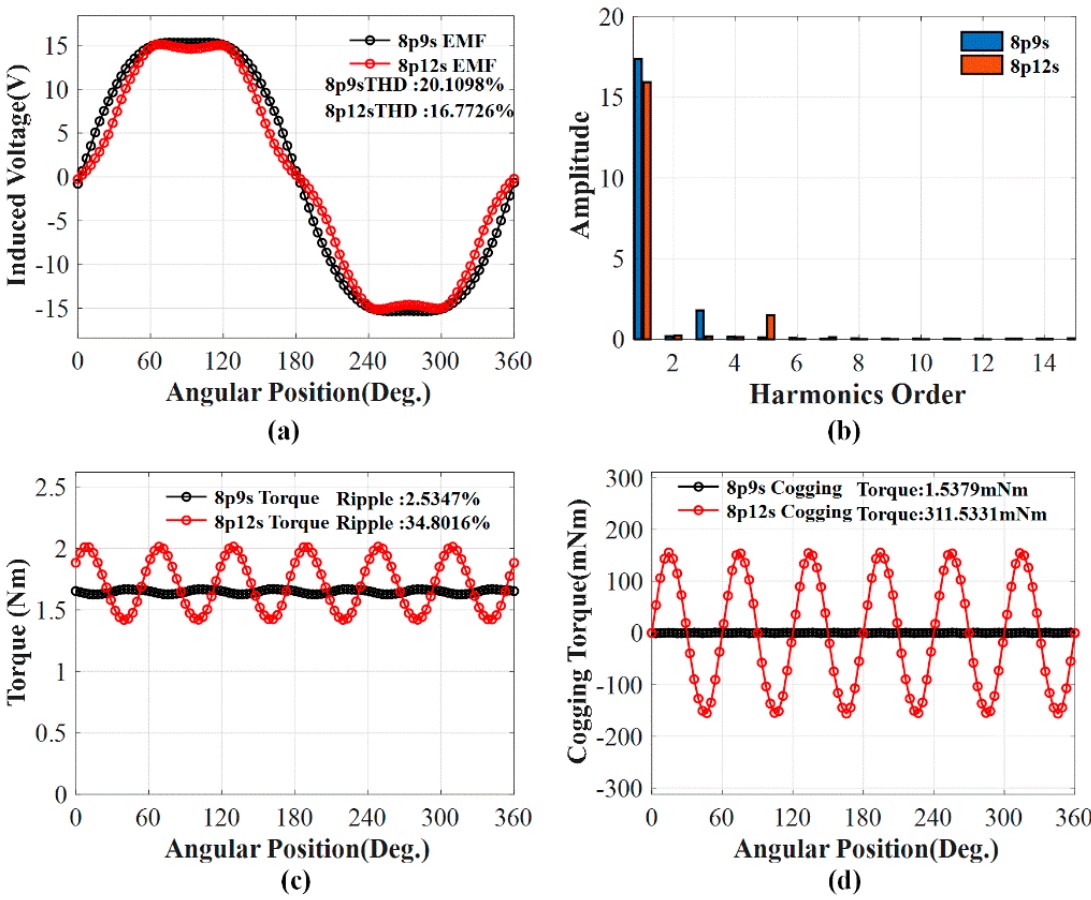

**Figure 2.** Comparison of the analysis results of permanent magnet machines with two different pole/slot combinations: (**a**) back-EMF, (**b**) THD of back-EMF, (**c**) torque ripple, and (**d**) cogging torque.

Previous studies have presented the goodness factor equation of a pole/slot combination, which is derived using the pole number, the slot number, and the lowest common multiple (LCM) of the pole and slot. This means that as the goodness factor increases, the cogging torque increases. The goodness factor equation is expressed in Equation (3). Here, $N_{LCM}$ is the LCM of the pole and slot numbers. $Q_s$ and $p$ indicate the number of the slot and the number of the pole pair, respectively.

According to Equation (3), the magnitude of the cogging torque of the 9-slot model is higher than that of the 12-slot model.

$$C_t = \frac{2pQ_s}{N_{LCM}} \tag{3}$$

Figure 2c compares the results of the torque ripple analysis, indicating that the 9-slot model has better torque ripple characteristics than that of the 12-slot model. Figure 2d compares the results of the cogging torque analysis, indicating that the 12-slot model has better cogging torque characteristics than that of the 9-slot model. The torque ripple values of the 9- and 12-slot models are 20.1% and 16%, respectively; the cogging torques of the 9- and 12-slot models are 1.5379 and 311.5311 mNm, respectively, and it can be confirmed that the torque and EMF characteristics of the 9-slot model are better than that of the 12-slot model.

## 2.2. MPF and UMF

The electromagnetic force or UMF generated by the motor can be derived using the Maxwell stress tensor equation. The electromagnetic force or UMF of the object placed in an electromagnetic field can be calculated by integrating the electromagnetic stress with the closed surface of the object. Electromagnetic stress is defined in Equation (4). These equations are used to obtain the radial and circumferential (tangential) components of MPF.

$$f_r = \frac{1}{\mu_0}\left(B_r^2 - B_\alpha^2\right), \ f_\theta = \frac{1}{2\mu_0}B_r B_\alpha \tag{4}$$

Here, $f_r$ is the radial force density, $f_a$ is the radial force density, $\mu_0$ is the permeability, $B_r$ is radial flux density, and $B_a$ is tangential flux density.

Figure 3a,b show the MPF analysis results, indicating that the 9-slot model has an unbalanced distribution. In general, the symmetry of the MPF distribution can be used to determine its influence on vibration.

UMF is derived by converting the radial and circumferential force density distributions into the *x*- and *y*-components of the force density distribution and integrating them into the air-gap, as follows:

$$\begin{aligned} F_x &= [rl_a/2\mu_0] \int_0^{2\pi} \left(B_r^2 - B_\alpha^2\right)\cos(\alpha) - 2B_r B_\alpha \sin(\alpha)d\alpha \\ F_y &= [rl_a/2\mu_0] \int_0^{2\pi} \left(B_r^2 - B_\alpha^2\right)\sin(\alpha) + 2B_r B_\alpha \cos(\alpha)d\alpha \end{aligned} \tag{5}$$

Here, *Fx* is the force along the *x*-axis, *Fy* is the force along the *y*-axis, r is the radius of the rotor, la is the axial-length of the machine, $\mu_0$ is the permeability, $B_r$ is the radial flux density, $B_a$ is the tangential flux density, and $\alpha$ is the rotor position. In addition, UMF is generated by the specific harmonic orders of magnetic flux density, which is $np \pm 1$. This can be confirmed by the results of the fast fourier transform (FFT) analysis.

Figure 4a,b show the results of the FFT analysis, indicating that the 9-slot model has specific harmonic orders. Figure 4c,d show the UMF results, indicating that the 9-slot model is larger than the 12-slot model. According to the comparative analysis results, the harmonics of the spatial flux density distribution increase the amplitude of the UMF; the UMF values of the 9- and 12-slot models are 72.42 and 0.08 N, respectively, and it can be confirmed that the MPF and UMF characteristics of the 12-slot model are better than that of the 9-slot model. It was confirmed that the 9-slot model had good torque characteristics and the 12-slot model had good force characteristics; thus, the electromagnetic–mechanical analysis can confirm which factors have a significant influence.

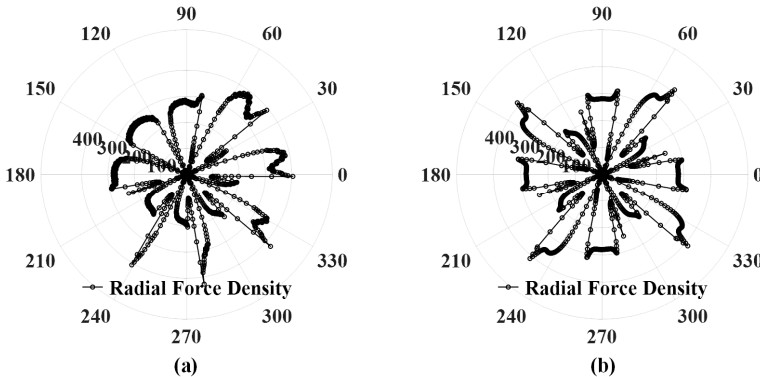

**Figure 3.** Comparison of the analysis results of permanent magnet machines with two different pole/slot combinations: (**a**) radial force density of 9-slots and (**b**) radial force density of 12-slots.

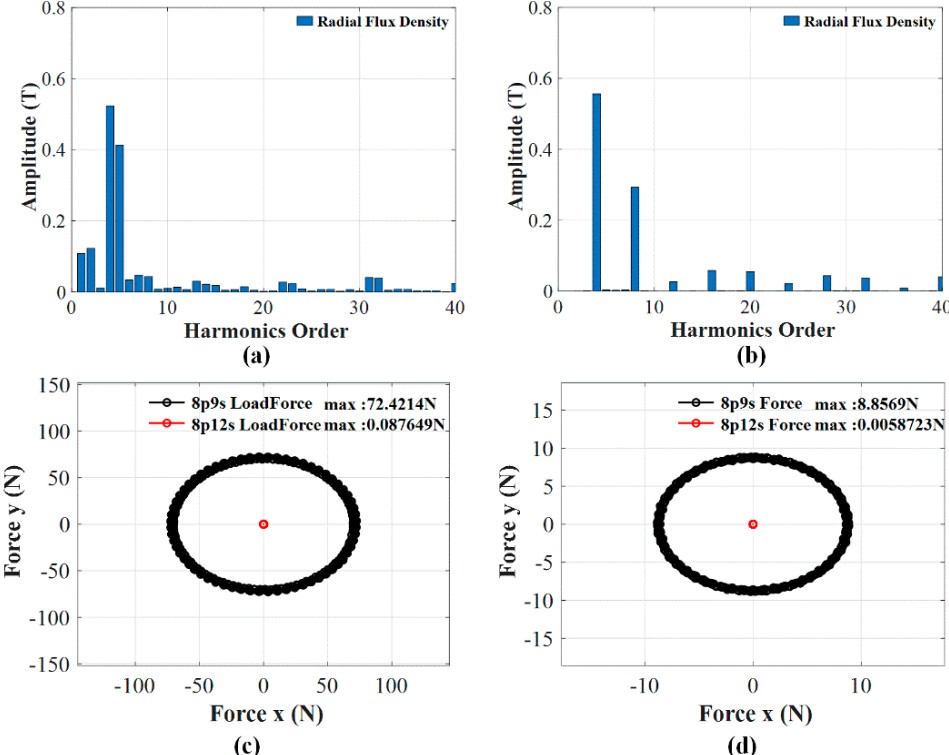

**Figure 4.** Comparison of the analysis results of permanent magnet machines with two different pole/slot combinations: (**a**) harmonics orders of radial flux density 9-slots, (**b**) harmonics orders of radial flux density 12-slots (**c**) UMF of no-load, and (**d**) UMF of rated-load.

## 3. Mechanical Analysis

### 3.1. Modal Analysis

In general, manufacturing defects such as the unbalanced mass and eccentricity of the rotor and bearing faults cause NVH (relatively low-frequency bands); thus, those are the major excitation frequencies in terms of NVH. In addition, vibrations caused by the electromagnetic sources may exist in major frequency bands, which can cause significant noise and resonance. Therefore, if the resonant frequency is close to the excitation frequency, the amplitude of NVH increases. Thus, it is important to confirm the natural frequency and mode shape, which are the mechanical characteristics of electrical machines, at the design stage. The stator of a rotational machine directly affects the electromagnetic vibration source and structurally influences NVH. It ensures that the resonance frequency exists in the

operation frequency band to exclude the effect of stator vibrations. Therefore, the natural frequency and mode shape of the stator should be analyzed to evaluate the NVH characteristics.

Figure 5a,b show the results of the modal analysis, indicating the natural frequency of each mode shape. The red part has the highest resonance value. The amplitudes (natural frequency) of the first, second, and third modes of each model are higher than those of the operation frequency. Thus, it is considered that the vibration based on the resonance of the structure of the stator will not occur.

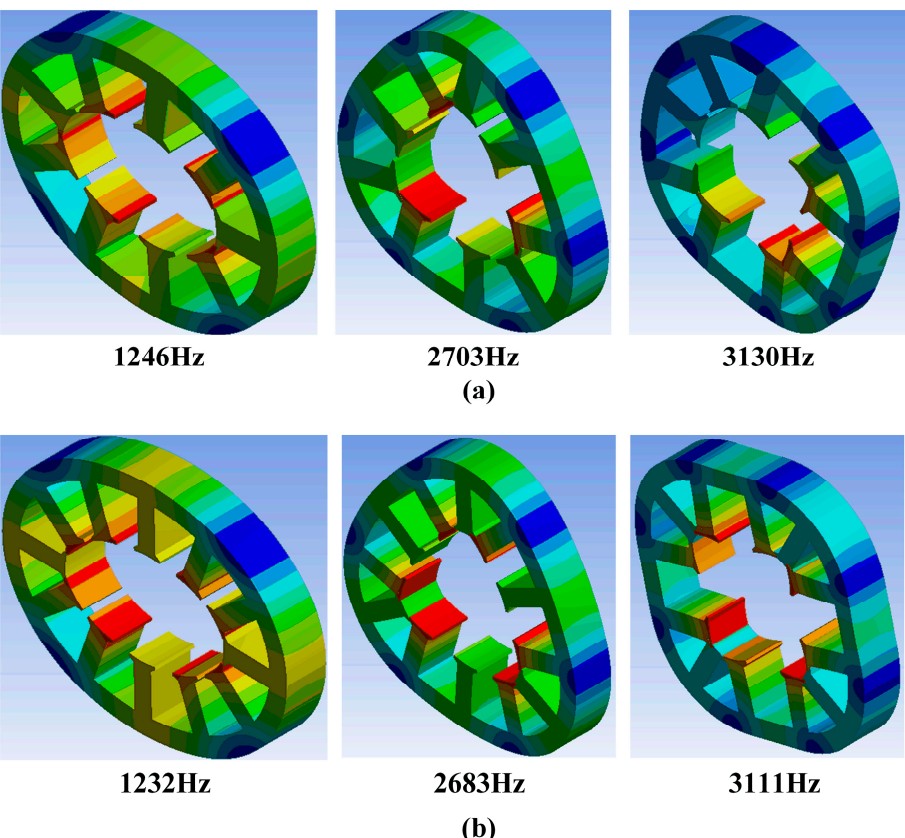

**Figure 5.** The modal analysis result of permanent magnet machines with two different pole/slot combinations: (**a**) 9-slot and (**b**) 12-slot.

### 3.2. NVH Analysis

To model the electromagnetic–mechanical analysis using the results of the electromagnetic analysis, it is necessary to set the boundary condition of the model and couple the electromagnetic analysis results. Based on the electromagnetic analysis model and results, a mechanical analysis was performed. In this study, the modal and forced vibration analyses were performed using ANSYS EM and WORKBENCH. The forced vibration analysis was performed by mapping the electromagnetic force with the air gap, which converted the harmonic orders generated by electromagnetic analysis using ANSYS EM and WORKBENCH, according to the 3D rotor model. Figure 6 shows the NVH analysis process of the PMSM in this study.

Figure 7a shows the displacement analysis results based on the center of the shaft axis using forced vibration analysis, indicating that the displacement of the 9-slot model is greater than that of the 12-slot model. Figure 7b,c show the results of the amplitude ratio and displacement analysis at rated load and no load. The red part has the highest resonance value. The analysis results show that the 9-slot model, with a large electric magnetic force, has greater deformation. The comparison of the electromechanical analysis results implies that the vibration of machines is electromagnetically influenced by MPF and UMF rather than torque pulsations.

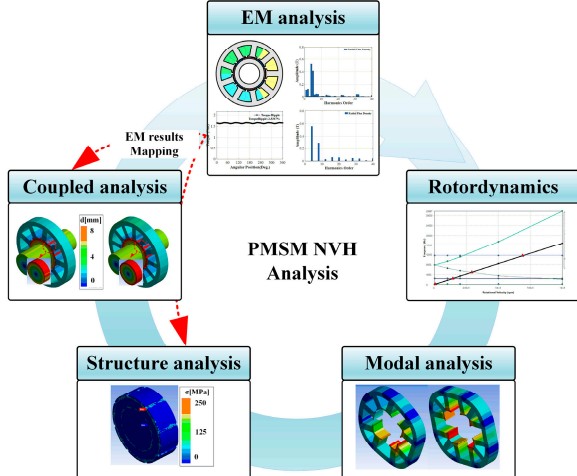

**Figure 6.** The noise, vibration, and harshness (NVH) analysis process of the permanent magnet synchronous motor (PMSM).

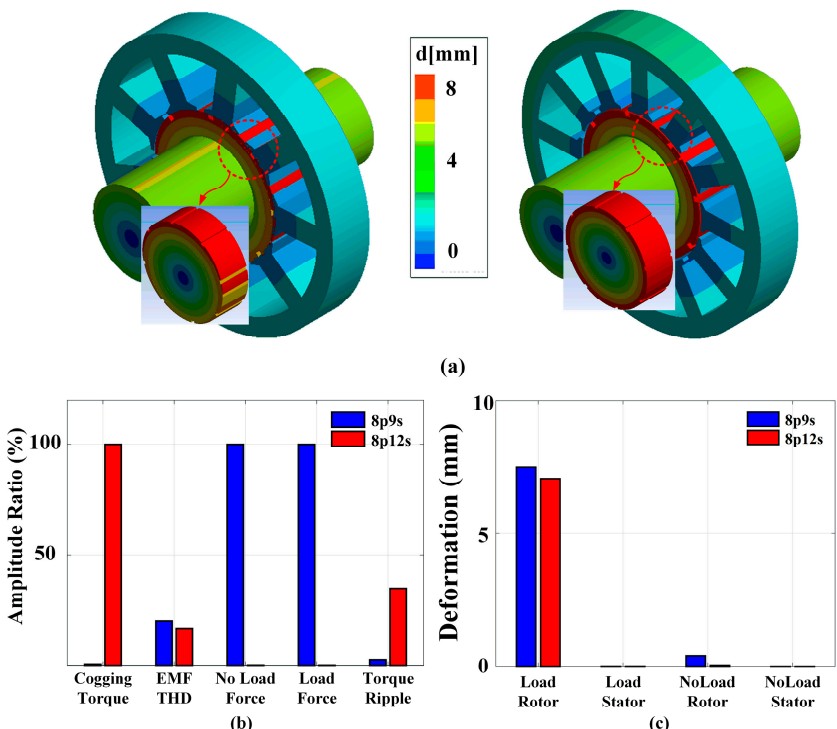

**Figure 7.** The analysis model of permanent magnet machines with two different pole/slot combinations: (**a**) electromagnetic analysis model and (**b**,**c**) prototype model.

Figure 8 shows the NVH characteristic analysis results of PM machines with two different pole/slot combinations by applying an electromagnetic vibration source. As shown in Figures 7 and 8, the NVH characteristics of the 9-slot model are more significant than that of the 12-slot model. Moreover, it can be seen through electromagnetic–mechanical analysis that vibration and noise occur at a higher level than the 12-slot model in the case of the 9-slot model in the overall harmonic order. Additionally, the 8th harmonic vibration component generated by the UMF is larger for the 9-slot model than the 12-slot model. The vibration amplitudes of the 9- and 12-slot models are 144 and 60 dB, respectively. Table 1 lists the comparative analysis results of PM machines with two different pole/slot combinations and an electromagnetic vibration source.

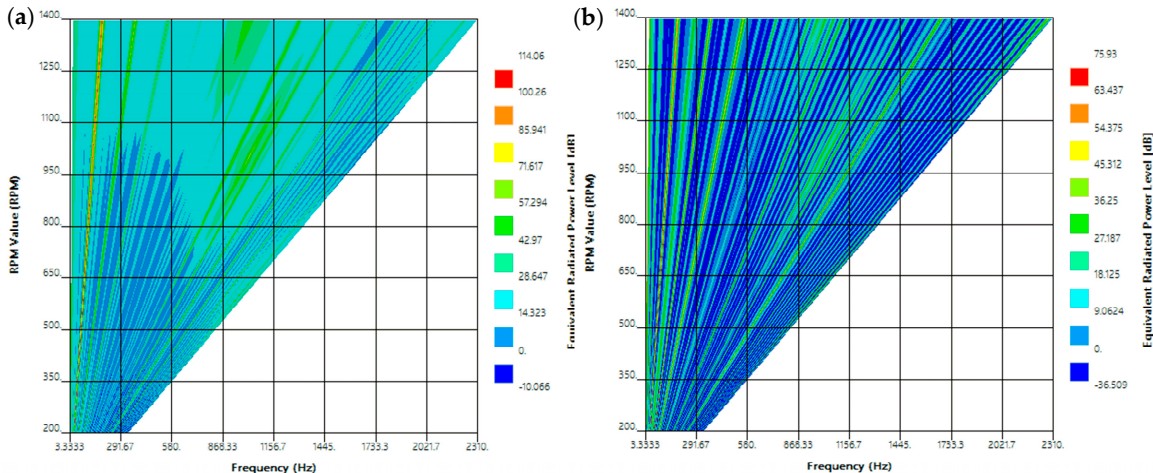

**Figure 8.** NVH characteristic analysis results of permanent magnet machines with two different pole/slot combinations: (**a**) 8-pole 12-slot and (**b**) 8-pole 9-slot.

**Table 1.** The comparison of results of PMSMs according to slot.

| Parameter | 9 Slot | 12 Slot | Harmonics Order |
|---|---|---|---|
| Back-EMF THD | 20.10% | 16.77% | 1,3,5 |
| Cogging torque | 1.53 mNm | 311.53 mNm | 6 |
| Torque ripple | 2.53% | 34.80% | 6 |
| No-load UMF | 8.85 N | 0.01 N | 8 |
| On-load UMF | 72.42 N | 0.08 N | 8 |
| Deformations | 2.5 um | 0.04 um | 1,3,4,6,7,8 |
| Main harmonics | 1,3,4,6,7,8 | 1,3,4,6,7 | - |

## 4. Experimental Results

To compare the analysis and experimental results with actual displacement measurements, a prototype PMSM model, including a rotor and a stator, is proposed. The displacement measurement concept is shown in Figure 9. The concept employs a gap sensor that measures shaft displacement. Figure 10 shows a system to measure vibration using two gap sensors. The displacements along the *x*- and *y*-axes were measured.

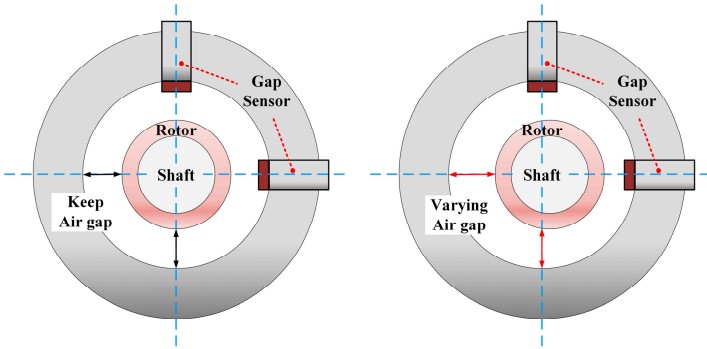

**Figure 9.** Schematic diagram of the displacement measurement concept.

The experimental results were validated by comparing them with the measurement results. The gap, torque, and speed sensors were connected to mechanically measure the displacement between the designed model and the sensor, torque, and speed of the continuous rotation of the designed model, namely, the PU-06A model of AEC, the CSA-1k model of CAS, and the MP-981 model of

ONOSOKKI. An eddy–current-type gap sensor was used so that it can measure a maximum length of 1 mm. A magneto-type speed sensor was used so that it can be inserted into the desired position. The adopted indicator is the SS-130 model of CAS, which can work with both models. The servomotor and invertor, namely, the L7SA model and the APM-FCL06A model of LS-mecapion, were connected to electrically operate at command rotational speed. The servomotor controller controls the motor with analog signals using a controller box configured using the supplied manual. It can be operated at maximum load conditions of 5000 rpm and 2 Nm. The servomotor was operated using a pulse-width modulation inverter with an analog signal control box, which can be operated based on user inputs. The designed model is operated by using an SPWM inverter that comprises an internal feedback loop. The rotational encoder is employed for position control.

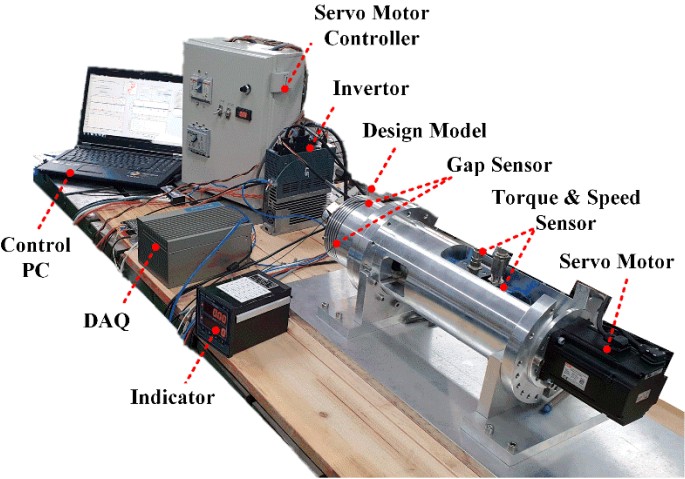

**Figure 10.** Vibration measurement system of the PMSM.

The experiment results are shown in Figure 11. It is observed that the displacement of the 9-slot model is 23% greater than that of the 12-slot model. In addition, comparing the harmonics of the displacement, the 9th-order machine produces an 8th-order harmonic caused by MPF and UMF, which affect the machines. This shows that MPF and UMF significantly influence the vibration of the motor. Table 2 lists the comparative results of the PM machines with two different pole/slot combinations.

Compared with the electromechanical analysis, the results of the analysis model are derived by applying the motor unit. Therefore, it derives vibrations from the motor unit, resulting in an error when compared with the vibration test results of the entire system. For accurate analysis, it is necessary to assess the material characteristics and shapes of parts applied to the system, such as servomotors and housings, and the characteristics of the electromagnetic excitation source generated by the servomotor; however, the purchased parts are difficult to apply. Moreover, as the tendency of vibration harmonics that occur frequently is constant, the proposed method is considered to be effective in analyzing the tendency of vibration generated by the motor.

**Table 2.** Deformation analysis of the results of PMSMs according to slot.

| Parameter | 9-Slot | | 12-Slot | |
|---|---|---|---|---|
| Classification | FEA | Experiment | FEA | Experiment |
| Deformations | 1.6 um | 2.5 um | 1.3 um | 0.04 um |
| Main Harmonics | 1,3,4,6,7,8 | 1,3,4,6,7,8 | 1,3,4,6,7 | 1,3,4,6,7 |

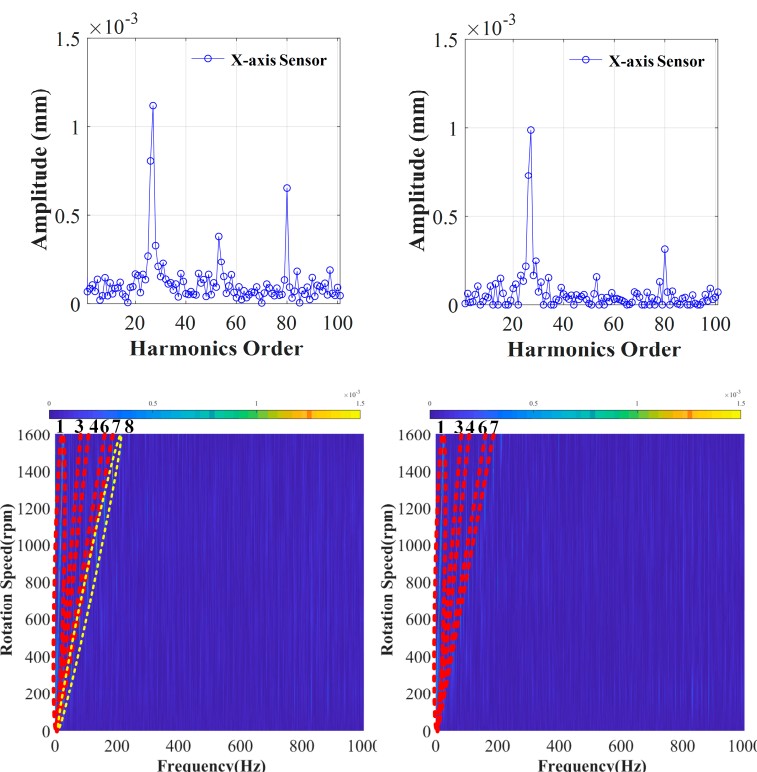

**Figure 11.** The displacement measurement results of permanent magnet machines with two different slots.

## 5. Conclusions

In this study, the electromagnetic vibration source of the PMSM was classified as follows: THD of the back-EMF, cogging torque, torque ripple, and UMF at the center of the air-gap. These were validated using coupled analysis and vibration experiments with prototype models. Characteristics analysis of each source was performed and then coupled analysis was performed. Torque pulsation and MPF were effective for the 9-slot model, and back-EMF and UMF were effective for the 12-slot model. According to mechanical analysis, the vibration characteristics were excellent for the 12-slot model. Based on the mechanical measurement, it was confirmed that the torque characteristic improved. In previous studies, it could be seen that the electromagnetic force at the center of the air gap, which differs in amplitude depending on the pole/slot combination, affected the vibration of the device. This may be disadvantageous to vibration and noise characteristics, even if the characteristics of UMF and MPF do not improve. Subsequently, the result of the analysis can be applied to the design process in order to improve vibration characteristics based on analysis methods.

**Author Contributions:** J.-Y.C.: conceptualization, review and editing; T.-K.B.: analysis, original draft preparation; K.-H.S.: experiment and co-simulation; H.-K.L.: experiment and motor control algorithm; J.-I.L.: co-simulation; H.-W.C.: conceptualization and analysis method review. All authors have read and agreed to the published version of the manuscript.

**Funding:** This work was supported by the Korea Institute of Energy Technology Evaluation and Planning (KETEP), the Ministry of Trade, Industry, and Energy (MOTIE) of the Republic of Korea. (no. 20183010025420), and the National Research Foundation of Korea (NRF) grant funded by the Korea government (MSIT; no. 2020R1A2C1007353).

**Conflicts of Interest:** The authors declare no conflict of interest.

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
