# Peer review of "Experimental and Comparative Study of Rotor Vibrations of Permanent Magnet Machines with Two Different Fractional Pole/Slot Combinations"

_applsci, doi:10.3390/app10248792_

Round 1

Reviewer 1 Report

The paper is dealing with vibration analysis of permanent magnet synchronous machines.

There are 14 references listed in the References chapter. However, in the text no references can be found although it is visible, that there are citations of someone else's work. Hence, this manuscript can be considered a draft, not a serious manuscript, intended to be accepted for publication. Using someone else's work without referring to it in the text can be considered plagiarism and must not be tolerated.

Additionally, there are unexplained abbreviations already in abstract, which is not acceptable. There is a large number of mistakes in text, as well as misuse of words and grammatical errors. Proof reading is essential for the manuscript. A number of figures is of low quality and must be improved.

Overall, as said, this is not a serious manuscript to be accepted. The authors have not taken the academic ethics seriously, hence, I find the paper must be rejected. However, I encourage the

Author Response

First of all, we appreciate your very generous and kind comments. Since the main objective of our paper is to develop NVH analysis and finding the NVH sources that have higher effect for permanent magnet machines. Therefore, based on your generous and kind comments, we revised our manuscript by adding some related references and paragragh. Even though we modified our manuscript, if it was not sufficient, we are always ready to revise and to improve it for its contribution to the related field.

Reviewer 2 Report

The paper presents experimental study of rotor vibration of PMSM for two pole/slot combinations.

Here are my comments:

  • Some of the figures are very small (e.g. Fig 2-4) and can be enlarged.
  • The introduction section is very short. The contributions of this paper is summarized in only one paragraph. This part can be expanded. Also the terms cogging torque, UMF, MPF needs to be explained further. 
  • The citation format is not consistent with the template.
  • Figure should appear in the paper after they are referred to. Placement of Figures 5, 6, 9, 10, and 11 needs to be changed.
  • Figure 5 needs more explanation including the significance of colors.
  • The components in Figure 10 need to be listed in detail. Are they built by the authors or are they purchased? Specifications of Invertor, servo controller, sensors, DAC, and indicator need to be provided.
  • Figures 7, 8, and 11 need further explanation and comparative analysis.
  • There are several grammar errors and typos in the paper. Comprehensive proofreading is recommended.
  • All the abbreviations should be elaborated the first time they appear in the text, e.g. NVH and EMF
  • Rare-earth magnet is listed as the first keyword while it never appears in the paper.
  • Lines 15 and 16 should be deleted.
  • The literature review seems not comprehensive. More recent or well-cited publications can be added.
  • The references should be formatted according to the template.
  • Are the simulation and experimental results compared? Can you provide a paragraph discussing the simulation vs. experimental results.
  • Can the results of the paper be compared with the improvements from the previous studies listed in the literature review? Please add a quantitative analysis.

Author Response

First of all, we appreciate your very generous and kind comments. Since the main objective of our paper is to develop NVH analysis and finding the NVH sources that have higher effect for permanent magnet machines. Therefore, based on your generous and kind comments, we revised our manuscript by adding some related references and paragraph. Even though we modified our manuscript, if it was not sufficient, we are always ready to revise and to improve it for its contribution to the related field.

Round 2

Reviewer 1 Report

The previous reviewer remarks have been sufficiently addressed.

There are still some issues with text editing and some of the Figs. might need zooming or better quality. I feel these are minor questions and can be solved prior to submitting the final version to be published.

Author Response

(The authors gave the same response as above.)

Reviewer 2 Report

I would like to thank the authors for their response and applying the suggestions. Here are my further comments:

1. The paper needs a comprehensive proofreading to resolve grammar and English errors. Parts of the paper is hard to understand or written unprofessionally. Please edit. Specially, the usage of article "the" in the paper needs a review.

2. Mass citation used in this paper is a very bad writing habit ([1-10] and [5-16]). In the literature review process, relevant documents need to be analyzed.

3. Results in most of the figures and tables in the paper are not discussed properly. A good technique to use for figures and tables is sandwiching where the figure or table is introduced in the text, and is fully discussed after the figure or table. Many of the figures and tables in the paper are not discussed at all. It is usually considered a bad habit to end a section with a figure or table.

4. Equation 4 is placed in bad part of the paper and is not explained either.

5. Do the results presented in Table II mean that simulation results are not reliable. More explanation is needed (including recommendations on how to improve the quality of simulation).

6. The abbreviation explanations in parts of the paper is not according to standards, e.g.: NVH (Noise, Vibration and Harshness) should be changed to Noise, Vibration and Harshness (NVH)

7. The equation fonts can be improved. 

Author Response

(The authors gave the same response as above.)
